# Cancer-Testis Antigens in Triple-Negative Breast Cancer: Role and Potential Utility in Clinical Practice

**DOI:** 10.3390/cancers13153875

**Published:** 2021-07-31

**Authors:** Runyi Adeline Lam, Tracy Zhijun Tien, Craig Ryan Joseph, Johnathan Xiande Lim, Aye Aye Thike, Jabed Iqbal, Puay Hoon Tan, Joe Poh Sheng Yeong

**Affiliations:** 1MD Programme, Duke-NUS Medical School, Singapore 169857, Singapore; e0415976@u.duke.nus.edu; 2Institute of Molecular and Cell Biology (IMCB), Agency of Science, Technology and Research (A*STAR), Singapore 138648, Singapore; tracytien96@gmail.com; 3School of Biological Sciences, Nanyang Technological University, Singapore 637551, Singapore; craig001@e.ntu.edu.sg; 4Division of Pathology, Singapore General Hospital, Singapore 169608, Singapore; johnathan.lim.xiande@sgh.com.sg (J.X.L.); daw.aye.aye.thike@sgh.com.sg (A.A.T.); jabed.iqbal@singhealth.com.sg (J.I.); tan.puay.hoon@singhealth.com.sg (P.H.T.); 5Singapore Immunology Network (SIgN), Agency of Science, Technology and Research (A*STAR), Singapore 138648, Singapore

**Keywords:** triple-negative breast cancer, cancer testis antigens, pathology

## Abstract

**Simple Summary:**

Triple-negative breast cancer (TNBC) has been associated with worse prognoses due to the limited treatment options. Thus, there is a need to characterise new biomarkers or treatment targets to improve patient outcomes. Cancer testis antigens (CTAs) are a group of antigens that are preferentially expressed in tumours and exhibit strong immunogenicity, as such, CTAs hold great promise as potential treatment targets and biomarkers in cancer. Previous reports have implicated roles for CTAs in different subtypes of breast cancer, including TNBC. Multiple clinical trials are in progress investigating CTAs as treatment targets in various cancers. This review aims to discuss the roles of CTAs in TNBC and discuss the potential applications and benefits of incorporating CTAs in clinical practice.

**Abstract:**

Breast cancer cells commonly express tumour-associated antigens that can induce immune responses to eradicate the tumour. Triple-negative breast cancer (TNBC) is a form of breast cancer lacking the expression of hormone receptors and cerbB2 (HER2) and tends to be more aggressive and associated with poorer prognoses due to the limited treatment options. Characterisation of biomarkers or treatment targets is thus of great significance in revealing additional therapeutic options. Cancer-testis antigens (CTAs) are tumour-associated antigens that have garnered strong attention as potential clinical biomarkers in targeted immunotherapy due to their cancer-restricted expressions and robust immunogenicity. Previous clinical studies reported that CTAs correlated with negative hormonal status, advanced tumour behaviour and a poor prognosis in a variety of cancers. Various studies also demonstrated the oncogenic potential of CTAs in cell proliferation by inhibiting cell death and inducing metastasis. Multiple clinical trials are in progress to evaluate the role of CTAs as treatment targets in various cancers. CTAs hold great promise as potential treatment targets and biomarkers in cancer, and further research could be conducted on elucidating the mechanism of actions of CTAs in breast cancer or combination therapy with other immune modulators. In the current review, we summarise the current understandings of CTAs in TNBC, addressing the role and utility of CTAs in TNBC, as well as discussing the potential applications and advantage of incorporating CTAs in clinical practise.

## 1. Introduction

Breast cancer is a heterogeneous disease that can be classified based on the clinical, morphological and biological characteristics [1,2]. Triple-negative breast cancers (TNBC), a special class of breast cancers that are negative for oestrogen receptor (ER), progesterone receptor (PR) and HER2 (cerbB2), represent 9–17% of all breast carcinomas, depending on the threshold for ER, PR and HER2 positivity [1,3,4,5]. TNBC are also of high histological grade, displaying aggressive clinical behaviour with poorer prognosis and accompanied by frequent metastasis to the brain and lungs, with a shorter time to recurrence and death [3,5]. 

Clinical outcomes in breast cancer based on disease-free survival (DFS) and overall survival (OS) have been enhanced over the years by incorporating different factors, such as molecular classification, as well as other clinical parameters, such as menopausal status, performance status and the stage of the disease, in treatment algorithms [5,6]. Despite the recent advances, issues such as the significant number of non-responders limit the effectiveness of personalised treatment. Results from a retrospective study involving 9156 breast cancer patients found that survival for guideline-adherent non-TNBC patients are significantly better compared to guideline-adherent TNBC patients, and the underlying reason is currently not known [7]. Thus, there is a constant need to improve the protocols and to formulate new strategies for human breast cancer management, especially for patients with limited treatment options such as those with TNBC [5]. Several studies have identified novel biomarkers and new strategies to improve the diagnosis and treatment of TNBC [3,8,9,10,11,12,13,14,15,16,17].

Cancer testis antigens (CTAs) are tumour antigens that are expressed normally in embryonic stem cells and testicular germ cells and minimally expressed in most other tissues. CTAs are aberrantly found in various cancers, especially advanced cancers with stem-cell-like characteristics [18,19]. CTAs were first identified in 1991 in a study showing that the presence of *MAGEA1* caused resistant tumour cell clones to be sensitised to killing by autologous cytotoxic T-lymphocytes [20]. Since then, there has been an explosion of CTA-related research and the discovery of more CTAs, including NY-ESO1, which is the most successful target to date for cancer immunotherapy [21,22]. The number of CTAs identified has also increased exponentially over the years. In 2009, a database was created of about 70 families and more than 200 members of CTAs [23]. With the advent of next-generation sequencing, there has been a huge increase in genomic data. By integrating transcriptomic data from multiple databases, Wang et al. systematically identified 876 new CTAs in 19 cancers [24], and a different research group found an additional 201 new CTAs [25].

Various studies have found the expression of individual or sets of CTAs in tumours correlates with worsening survival, thus there has been speculation that CTAs could promote aggressive tumour growth or increase the resistance of cancer to chemotherapy [26,27,28,29]. Several studies have also described the oncogenic role of individual CTAs in human tumour cells, and the CTA functions can be divided into three categories: transcriptional regulation, mitotic accuracy, and protein turnover [29]. CTAs’ cancer-restricted pattern of expression, strong immunogenicity and oncogenic roles in cancer make them attractive targets for cancer immunotherapy and the development of CTA-based vaccine treatments for patients with limited treatment options, such as those with TNBC. 

In this review, we discuss the role and potential applications of CTAs in TNBC. Based on our understanding, this is the first article to comprehensively review CTAs that are expressed in TNBC.

## 2. Expression of CTAs in Triple-Negative Breast Cancer

TNBC frequently express CTAs (Figure 1), and a summary of the CTAs commonly associated with TNBC is shown in Table 1.

### 2.1. CTAs Associated with Worse Outcome in TNBC

Several CTAs are frequently overexpressed in TNBC and have shown correlation with poorer patient outcomes (Figure 1, Table 1). One prominent example is the melanoma antigen gene (MAGE) family, which has over 60 proteins divided into Type I (MAGE-A, B and C subfamilies) or Type II (MAGE-D, -E, -F, -G, -H, -L and necdin) depending on the positions of their genes in relation to the X-chromosome [70,71,72].

MAGE-A is frequently expressed in solid tumours such as non-small cell lung cancer (NSCLC), bladder cancer, oesophageal and head and neck cancers, sarcomas and TNBC [73]. A meta-analysis of 7428 patients and 44 studies found that high MAGE-A expression was significantly correlated with worst outcomes for various solid cancers such as lung, gastrointestinal, breast and ovarian cancers [71]. In addition, several groups also found higher expression of MAGE-A and NY-ESO-1 in ER-negative versus ER-positive breast cancers [33,34]. Immunohistochemistry (IHC) analysis of eight CT antigens, including MAGE-A and NY-ESO-1, also showed remarkably higher expression in ER-negative versus ER-positive human breast cancers [35]. Higher expression of MAGE-A was reported to define a very aggressive subtype of TNBC and to be associated with poor outcomes [36]. In other studies, MAGE-A3, -A6 and -C2 expression levels in breast cancers were significantly correlated with parameters such as ER-negative or PR-negative status, tumour grade and outcome, while MAGE-A10 expression was associated with TNBC status [37,38,39].

Interestingly, the expression of MAGE was higher compared to that of NY-ESO-1 in TNBC, with values ranging from 22% to 93% (Table 1). Curigliano et al. observed higher MAGE-A levels in TNBC (32%) compared to ER-positive breast cancer samples (10%) [33]. The findings were corroborated by another study which reported higher MAGE-A expression in TNBC (85.7%) compared to ER^+^PR^+^HER2^−^ breast cancer samples (55.5%) [39].

The mesothelin (MSLN) gene encodes a glycosylphosphatidylinositol (GPI)-anchored membrane protein, which gives rise to MSLN and a soluble product, known as megakaryocyte potentiating factor, upon cleavage [74]. MSLN is expressed on the membrane of mesothelial cells of the peritoneal and pleural cavities, and overexpression of MSLN has been observed in mesothelioma, ovarian, lung, oesophageal, endometrial and breast cancers [75]. MSLN was significantly expressed in TNBC compared to non-TNBC tissues and was an independent prognostic marker associated with distant metastasis and worse survival (Figure 1, Table 1) [47].

Prostate stem cell antigen (PSCA) is a member of the Thy-1/Ly-6 family that encodes a 123-amino acid GPI-anchored cell surface protein (Table 1) [49]. Upregulated PSCA expression was found in breast cancer patients and was associated with unfavourable histopathological grade, increased Ki67 proliferation index and HER2/neu receptor status [49]. Patients with PSCA-positive invasive micropapillary carcinoma (IMPC) of the breast had decreased DFS compared to PSCA-negative IMPC patients [50].

Receptor tyrosine kinase-like orphan receptor 2 (ROR2), a novel Wnt receptor, belongs to the tyrosine kinase receptor family, which is important in regulating skeletal and neuronal development, cell migration and cell polarity (Table 1) [76]. Breast cancer patients, including those with TNBC expressing ROR2, experienced a significantly worse prognosis with shorter overall survival compared to those lacking ROR2 [51].

Sperm protein associated with the nucleus X-linked (SPANX) is a CTA with a role in spermatogenesis (Table 1) [77]. High nuclear and cytoplasmic SPANXB1 expression patterns were observed in 73% (11/15) of human primary and metastatic TNBC tissues in contrast to normal tissues [52]. SPANX-A/C/D enhanced the metastasis of ER-negative breast cancer cells, and high SPANX-A/C/D levels in breast cancer tumours correlated with poor prognosis, with increasing SPANX-A/C/D levels being associated with a shorter distant metastasis-free survival time in ER-negative patients [53].

A-kinase anchoring proteins (AKAP) are a group of CTAs involved in sperm function (Table 1) [78]. AKAP3 expression was deficient in TNBC, and patients with AKAP3-positive TNBC have a better 5-year DFS [30].

### 2.2. CTAs Associated with Improved Outcome in TNBC

The CTA New York oesophageal squamous cell carcinoma-1 (NY-ESO-1), also known as cancer-testis antigen 1B (CTAG1B), is immunogenic and reportedly induces specific B-cell and T-cell immunity in patients with NY-ESO-1-expressing cancers [79]. NY-ESO-1 was not frequently expressed in non-TNBC, with expression levels ranging from 0% to 12.5% [33,35,39,40,43,54,56,57]. In contrast, expression levels of NY-ESO-1 in TNBC ranged from 9.3% to 28.6% (Table 1). NY-ESO-1 was preferentially detected in TNBC compared with ER-positive tumours, with NY-ESO-1 expression reported in nine (18%) TNBC compared to two (4%) ER-positive tumours [33]. Consistent with these findings, a separate study noted the expression of NY-ESO-1 in 16% of TNBC compared with only 2% in ER-positive tumours, and 8 out of 11 (72.7%) patients with NY-ESO-1+ TNBC had measurable antibody responses to NY-ESO-1 [54]. NY-ESO-1 levels correlated with the number of tumour-infiltrating lymphocytes (TILs) and were also associated with good prognosis in 1234 TNBC samples [56]. Taken together, the study findings suggest that NY-ESO-1 is commonly expressed at high levels in TNBC and TBNCs expressing higher levels of NY-ESO-1 tend to have a better prognosis than TNBCs with lower levels of expression.

### 2.3. CTAs with Oncogenic Potential

CTAs have been involved in various processes of tumourigenesis and metastasis, such as epithelial-mesenchymal transition (EMT), proliferation, apoptosis and invasion (Table 1). MAGE-A has been implicated in highly aggressive behaviour and EMT in TNBC [42]. Analysis of 120 female breast cancer patients who underwent mastectomy revealed higher MAGE-A staining in TNBC (76.47%, 13/17), which was significantly correlated with ER-negative, PR-negative and HER-2-negative statuses, lymph node involvement and higher histological grade [42]. A reduction in MAGE-A was associated with increases in epithelial markers and declines in mesenchymal characteristics [42]. The biological functions of MAGE-A family members have not been fully elucidated, but various reports suggest that MAGE proteins can promote tumourigenesis and metastasis via different mechanisms, such as acting as a master regulator of E3 RING ubiquitin ligase, by inhibiting p53 tumour suppressor or by enhancing cell motility [80,81]. Similarly, very little is known about the biological functions of NY-ESO-1, with some reports suggesting that it might be involved in cell cycle progression and growth [21].

Preferentially expressed antigen of melanoma (PRAME) encodes a membrane-bound protein that promotes autologous cytotoxic T-cell-mediated immune responses (Table 1) [82]. An analysis of 295 breast cancer patients revealed that higher expression of PRAME was associated with poorly differentiated tumours, and PRAME expression was related to ER levels, as ER-negative patients had higher PRAME expression, whereas ER-positive patients had low PRAME expression [58]. PRAME expression was reported to be higher in basal-like breast cancer subtypes than other subtypes [59]. In addition, PRAME reportedly plays a role in tumourigenesis in TNBC through EMT-related gene reprogramming [60].

SPANX-A/C/D was discovered as an important factor for the metastasis of breast cancer cells; it was hypothesised to act via interactions with components of the cytoskeleton at the inner nuclear membrane and is required to produce actin-rich cellular protrusions during modelling of the extracellular matrix (Table 1). Sperm-associated antigen 9 (SPAG9) mRNA and protein expression was found in the cytoplasm of all examined breast cancer cells, including TNBC cells (Table 1) [62]. Interestingly, the downregulation of SPAG9 ameliorated the invasiveness of TNBC [62].

Zinc-finger protein 165 (ZNF165) is a member of the SCAN-(C_2_H_2_)_n_ sub-family of zinc-finger proteins and is predominantly found in the nucleus where it interacts with proteins involved in gene-regulatory activity (Table 1) [27]. ZNF165 is frequently overexpressed in TNBC and interacts with SMAD3 to promote the mitosis and survival of human TNBC cells by controlling the transcription of transforming growth factor β (TGFβ)-dependent genes both in vitro and in vivo [65]. The tripartite motif-containing 27 (*TRIM27*) gene encodes the RING finger protein (RFP), a member of the tripartite motif family that is involved in gene regulation [65]. TRIM27 was found to be critical for the transcriptional activity of ZNF165 in promoting tumour growth in TNBC. Higher expression of *TRIM27* was reported in TNBC compared to normal breast tissue using data from The Cancer Genome Atlas (TCGA) [65]. Interestingly, TRIM27 exhibited different subcellular localisation with transformation, with higher nuclear expression in TNBC, suggesting that the transcriptional activity of TRIM27 might be associated with the transformed state [65].

Testes-specific protease 50 (*TSP50*) is an oncogene that promotes breast cancer survival, invasion and metastasis via the activation of the NF-κB signalling pathway (Table 1) [6]. Knockdown of *TSP50* in breast cancer cells significantly inhibited cellular proliferation [64]. The levels of TSP50 together with the expression of p65 and matrix metalloproteinase 9 (MMP9) were analysed in conjunction with clinicopathological features, such as tumour size, pathologic grade, ER and PR levels, in breast cancer tissues [63], and the majority of TSP50+/ p65+ tumours (72%) and TSP50+/MMP9+(78%) tumours were negative for ER expression and tended to be of a higher grade [63].

### 2.4. CTAs with Increased Expression in TNBC but with Unclear Implications

Some CTAs display increased expression in TNBC, but their significance has not been clearly defined. In large-scale in silico analyses, actin-like 8 (ACTL8), which is the building block for the intracellular architecture of cells, was expressed in 57% of TNBC, while Kita-Kyushu lung cancer antigen-1 (KK-LC-1), also known as cancer-testis antigen 83 (CT83), chromosome X open reading frame 6 (*CXorf6*), or mastermind-like domain containing 1 (*MAMLD1*), was expressed in 65% of TNBC [66]. KK-LC-1/CT83/ *CXorf6/MAMLD1* was found to be a critical gene in hypospadias and plays a less important role in testosterone production during the critical time for sex development [83,84]. KK-LC-1 was found in all TNBC tumours and all tumours without ER expression, with gene and protein expressions of 11.8% and 52.9%, respectively [68]. In another study, it was found that 53% of TNBC patients expressed *CXorf61* in at least 30% of their tumour cells, whereas expression was strictly restricted to the testes in normal human tissues (Table 1) [67]. *CXorf61* has the potential to induce a strong immune response, as high frequencies of CXorf61-specific T cells could be obtained by vaccinating HLA-A*02-transgenic mice with *CXorf61*-encoding RNA, and in vitro priming of human CD8+ T cells derived from a healthy donor recognizing CXorf6166-74 induced a strong antigen-specific immune response [67].

Sperm protein 17 (SP17) is involved in various stages of spermiogenesis, and aberrant SP17 expression has been linked to cancers such as ovarian, oesophagus, central nervous system, multiple myeloma and esthesioneuroblastoma (Table 1) [85,86,87,88,89,90]. The exact role of SP17 in cancer cells has not yet been elucidated, with some reports suggesting that SP17 promotes cell–cell adhesion in malignant B-lymphocytes via interaction with heparan-sulphate and enhances cell movement and drug resistance in ovarian cells [91,92]. SP17 was preferentially expressed in breast cancer cell lines and primary breast tumours, including TNBC, compared to non-tumoural breast tissue [69]. Specific anti-SP17 antibodies were also discovered in patients’ sera, and the generation of SP17-specific, HLA class I-restricted, cytotoxic T-lymphocytes led to the death of breast cancer cells, opening the possibility that SP17 could be a valid target for TNBC immunotherapy [69].

Wilms tumour antigen (WT-1) is a transcription factor containing four zinc-finger motifs at the C-terminus and a proline or glutamine-rich DNA-binding domain at the N-terminus [93]. Out of all breast cancer subtypes examined, TNBC had the highest WT1 expression, with WT1 expression (score > 2+) being recorded in 27 (54%) TNBC cases, but only 6 (12%) and 3 (6%) of ER-positive and HER2-positive tumour cases, respectively (Table 1) [57].

## 3. Future Potential Application of CTAs Clinically

### 3.1. Future Potential Application of CTAs in Screening Workflow in Clinical Practice

As summarised in Table 1, majority of laboratories evaluated CTAs expressions in breast tissues with IHC platform using commercially available antibodies. It is possible to develop assays in the clinical settings using these antibodies. Conventional IHC is cost- effective, easy to execute and frequently performed to visualise expression patterns in human biopsy samples. However, there are limitations to the number of markers that can be evaluated in each tissue sample. Several technologies such as multiplex IHC/immunofluorescence(mIHC/IF), imaging mass cytometry, multiplex ion beam imaging, single-cell RNA sequencing have emerged to circumvent the limitations. Extensive reviews on these emerging technologies have also been discussed [94,95,96].

Multiple laboratories have utilised mIHC/IF to evaluate biomarkers in several cancers. Systematic review and meta-analysis of tumour patients from 8135 patients revealed that mIHC/IF had superior diagnostic accuracy in predicting clinical response to anti-PD-1/PD-L1 therapy than PD-L1 IHC, tumour mutational burden or gene expression profiling [97]. An optimised mIHC/IF protocol was developed for PD-L1 testing in TNBC which demonstrated good correlation with conventional IHC, thus providing further evidence for the feasibility of incorporating mIHC/IF in clinical practice [98]. A recent multisite study comparing mIF on PD-1/PD-L1 axis on tonsil and breast carcinoma and non-small cell lung cancer (NSCLC) demonstrated good reproducibility and sensitivity across multiple institutions which included Johns Hopkins University, Yale University, MD Anderson Cancer Center, Earle A. Chiles Research Institute, Akoya Biosciences and Bristol-Myers Squibb [99]. Standardisation, validation and reproducibility of end-to-end workflow across multiple sites and clinical laboratory processes are important to promote translation of mIHC/IF technology to clinical practice. The evidence from feasibility studies have been encouraging, and more research could be done in this area to translate emerging technologies to clinical practice. Recently, a high throughput system to analyse tissue microarrays of breast cancer samples using multiplexed microfluidic IHC to generate biomarker barcodes have been developed [100]. The biomarker barcode of breast cancer patient-derived tissue microarrays was compared to traditional method of breast cancer diagnosis, thus opening the possibility of high-throughput screening with diagnostic capability.

Advances in next-generation sequencing (NGS) technology have identified unrecognised gene expressions and somatic mutations in TNBC samples, thus allowing refinement of TNBC subtypes at the genomic level and the possibility of identifying novel targets for personalised therapy [101]. Using an expanded 358-gene NGS assay, JAX-CTP™ assay on the Illumina HiSeq 2500 or MiSeq sequencers, additional clinically relevant genetic variants were identified in TNBC, thus enlarging the possibility for additional therapeutic interventions and clinical trial eligibility for these patients [102]. To investigate the clinical value of whole genome sequencing (WGS) in improving TNBC patient stratification, 254 TNBCs were sequenced under the Sweden Cancerome Analysis Network-Breast (SCAN-B) study [103]. Homologous Recombination Deficiency Detect (HRDetect) mutational-signature-based algorithm was used to classify tumours, with HRDetect-high patients having better invasive disease-free survival and distant relapse-free interval on adjuvant chemotherapy compared to HRDetect-low and HRDetect-intermediate having the poorest outcome [103]. There are several advantages in utilising WGS or NGS in routine diagnostic settings such as better TNBC patient stratification by identifying different groups with different survival likelihoods compared to classification by individual mutations, identifying poorer responders to current standard of care that are not identified by other method, identifying tumours that do not have genetic/epigenetic drivers but might have good outcome.

Current evidence suggests that CTAs are a common feature of TNBC and may correlate with prognosis of patients with TNBC. At present, TNBC samples are not routinely screened for CTAs expressions. The expression and characteristics of CTAs in TNBC provide exciting possibilities to enhance current screening, diagnosis and treatment methods (Figure 2). We propose utilising conventional IHC or multiplex staining platform to evaluate CTAs in evaluation of TNBC samples. Genomics study using microarrays or NGS could also be applied to identify new or unique CTAs or molecular signatures in TNBC for better patient stratifications. We propose that application of enhanced screening in clinical practise would lead to quicker diagnosis, early detection of cancer, better patient stratification and identification of good or bad responders to treatments. The overall goal is to contribute to better treatment outcome and personalised medical therapy for patients with TNBC. This is important as TNBC is a heterogenous cancer with limited treatment options. While technologies such as mIHC/IF or NGS/WGS hold great promise to be translated to clinical practice, substantial barriers such as cost, technical issues and development of reliable and reproducible quality assurance and quality control prevent widespread adoption in clinical practice. While there is growing body of evidence applying these technologies clinically, more effort could be performed for widespread adoption and to take advantage of the technologies in clinical practice. A group of key opinion leaders and relevant stakeholders in Italy proposed guidelines on the use of NGS tests in clinical practice based on several parameters such as tumour types, number and complexity of biomarkers and availability of treatments [104]. Recommendations for technologies such as mIHC/mIF in clinical practice encompassing views from different expert could be collected for different cancers, encouraging growth in precision medicine.

### 3.2. Therapeutic Application of CTAs

CTAs such as NY-ESO-1, MAGE, MSLN, SPANXB1, PRAME, ZNF165, TRIM27, KK-LC-1, SP17 and WT-1 were found to be enriched in TNBC compared to non-TNBC tissue (Table 1). The specific enrichment of CTAs opens the possibility of CTAs as targets for personalised treatments or for identifying subtypes of patients with better or worse prognosis. In addition, CTAs are immunogenic, thus providing opportunities for development of therapeutic vaccines and are attractive targets for immunotherapy. Given the growing evidence of their roles in TNBC, CTAs are potential targets for therapeutic intervention. Several clinical trials are currently being conducted to evaluate the effectiveness of CTAs, such as NY-ESO-1, MAGE, MSLN, PRAME, PSCA, ROR2 and WT1, as treatment targets in breast cancers and other solid tumours (Table 2). CTAs are being widely developed as cancer vaccine, T-cell immunotherapy and antibody-based therapy in breast cancer including TNBC (Table 2). Other CTAs are also evaluated in clinical trials as single therapy or combination therapy in other cancer types, highlighting the attractiveness of CTAs as potential therapeutic strategy for cancers [105].

Some trials involving NY-ESO-1 showed promising results. LV305, a third-generation lentivirus-based vector, enhances expression of NY-ESO-1 in dendritic cells, and is tested in clinical trial against several tumour types such as sarcoma, ovarian, melanoma and lung cancers [106]. Induction of antigen-specific responses were observed in more than half of sarcoma patients and was associated with improved 1-year survival [106]. G305, a recombinant NY-ESO-1 protein vaccine with glucopyranosyl lipid A (GLA), a synthetic TLR4 agonist adjuvant, in a stable emulsion (SE) could stimulate NY-ESO-1-specific antibody in 75% of patients with solid tumours expressing NY-ESO-1 and T-cell responses in 44.4% of patients in a phase-1 study [107]. The cancer types in this study included melanoma, ovarian cancer, synovial sarcoma, NSCLC and breast cancer. To further boost the T-cells responses in patients, combination product of LV305 and G305, CMB305, is under investigation in Phase I (NCT02387125) and Phase II (NCT02609984) clinical trials and in combination therapy with immune checkpoint blockers [108]. Vaccination with the long synthetic NY-ESO-179–108 peptide with a strong immune adjuvant generated substantial and long lasting CD8+ and CD4+ T-cell responses lasting at least one year in stage III/IV melanoma patients, permitting further development of this vaccine formulation for cancer immunotherapy [109].

Members of the MAGE-A family have been developed for T-cell therapy with favourable results. In a Phase I study involving HLA-DPB1*0401-restricted T-cell receptors (TCRs) that was designed to specifically recognise MAGE-A3 antigen, complete remission was seen in a patient with metastatic cervical cancer, and partial remission was seen in 3 patients [110]. The adoptive T-cell therapy ADP-A2M4, which has been modified to express TCR that can recognise MAGE-A4 cancer antigen, achieved responses in nine tumour types (NCT03132922) [111].

### 3.3. Future Research Development

At present, there is limited information on the effect of race or ethnicity on the expression of CTAs in TNBC. NY-ESO-1 and MAGE are the most studied, with several reports involving TNBC patient samples from different continents, races and ethnicities. However, there are currently no reports directly comparing the impact of race and ethnicity on CTAs in TNBC. Analyses of *CTA1A* and *CTA1B* expression in lung cancer using data from TCGA database showed higher expression of these CTAs in Asians compared to non-Asians [112]. More studies should be conducted to address these questions of TNBC epidemiology.

Further research could be performed to analyse expressions of CTAs with different immune infiltrates and correlate with clinicopathological characteristics, given the growing evidence of CTA having immunogenic and oncogenic properties in TNBC and promoting tumourigenesis or invasiveness (Table 1). Several CTAs are co-expressed in TNBC, and further research could be performed to delineate different CTAs molecular signatures and its prognostic value for TNBC clinically. More research could be performed to understand mechanistically the underlying biological and cellular functions of CTAs in TNBC, as well as the relationships between CTAs and hormone receptor or HER2 signalling pathways. Current evidence suggests that some CTAs reduced the effectiveness of responses to chemotherapy, and a detailed understanding of the important signalling pathways in TNBC could help in designing more efficacious drugs.

A key aspect for investigation is whether CTAs are abnormally expressed in breast tissue before malignancy occurs and if they could be used as potential biomarkers for TNBC. This would help clinicians to identify women at high risk for cancer development, thus improving screening protocols for the early detection of TNBC (Figure 2).

## 4. Conclusions

TNBC frequently expresses CTAs, and some CTA expression correlates with overall survival and prognosis. CTAs show biased expression in cancer and robust immunogenicity, thus serving as ideal targets for cancer immunotherapy. Multiple clinical trials have been conducted or are currently on-going to investigate the role of CTAs as treatment targets in advanced cancers, such as TBNC. Further research could be conducted to delineate the mechanism of action of CTAs in TNBC, increasing the efficacy of CTAs or in combination with other immunotherapies, identifying patients that would benefit most from the treatment and devising better drug delivery. With the collation of more data, CTAs may also be incorporated in routine screening protocols for TNBC.

## Figures and Tables

**Figure 1 cancers-13-03875-f001:**
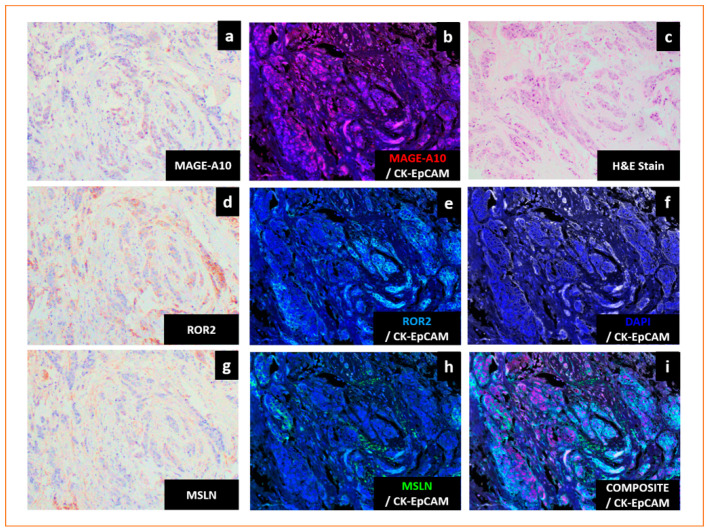
Cancer testis antigens are expressed in triple-negative breast cancers (TNBC). (**a**–**i**) Representative images of TNBC expressing melanoma antigen gene-A10 (MAGE-A10), receptor tyrosine kinase-like orphan receptor 2 (ROR2) and mesothelin (MSLN). (**a**) Immunohistochemistry (IHC) nuclear staining of MAGE-A10 (brown). (**b**) Multiplex IHC (mIHC) labelling of MAGE-A10 (red), cytokeratin- epithelial cell adhesion molecule (CK-EpCAM) (white) and DAPI (blue). (**c**) Haematoxylin and eosin (H&E) staining. (**d**) IHC membrane staining of ROR2 (brown). (**e**) mIHC labelling of ROR2 (cyan), CK-EpCAM (white) and DAPI (blue). (**f**) mIHC labelling of CK-EpCAM (white) and DAPI (blue). (**g**) IHC membrane staining of MSLN (brown). (**h**) mIHC labelling of MSLN (green), CK-EpCAM (white) and DAPI (blue). (**i**) mIHC labelling of MSLN (green), MAGE-A10 (red), ROR2 (cyan), CK-EpCAM (white) and DAPI (blue). H&E, IHC and mIHC were used to stain serial sections. Magnification: 200×.

**Figure 2 cancers-13-03875-f002:**
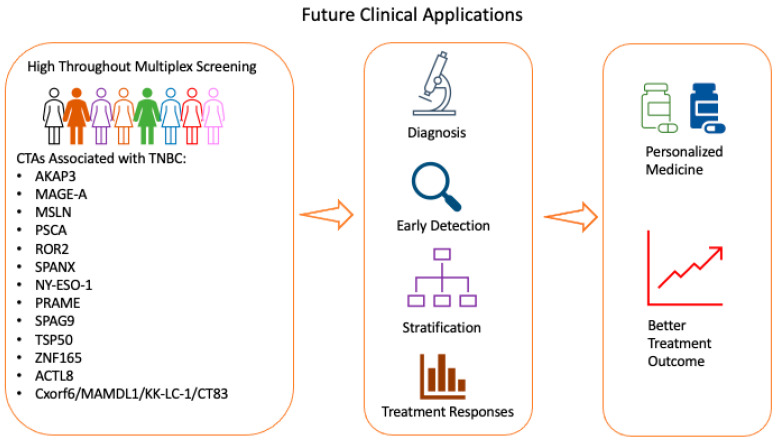
Potential workflow in screening for CTAs in TNBC. The expression of CTAs in TNBC provides great promise for enhancing current screening, diagnosis and treatment methods. TNBC: triple-negative breast cancer; CTA: cancer-testis antigens.

**Table 1 cancers-13-03875-t001:** Summary of CTAs in TNBC.

CTAs	Cellular Function	Institute	Cohort	Prevalence of CTAs in TNBC	Type of Assay	Antibodies	Role in TNBC	Ref.
CTAs associated with worse prognosis in TNBC								
A-kinase anchoring proteins (AKAP3)	Sperm function	Breast Cancer Research Centre (Tehran, Iran) [30]	Asian	20% (*n* = 25)	Real-Time Polymerase Chain Reaction (RT-PCR)		Loss of expression in TNBC. Breast cancer patients who were positive for AKAP3 had better 5-year disease-free survival.	[30,31]
Melanoma antigen gene (MAGE)	Not known. May promote tumourigenesis and metastasis.	Italian National Cancer Institute [32]	Caucasian	MAGE-A: 23% (*n* = 44)	IHC	MAGE-A Antibody (6C1)	Frequently overexpressed in TNBC. Higher expression of MAGE-A was reported to define a very aggressive subtype of TNBC and correlated with poor prognosis of patients. MAGE-A3, -A6 and -C2 expression in breast cancers was significantly associated with negative ER or negative PR status, higher-grade tumours and correlated with worse outcomes. MAGE-A10 expression was associated with ER-negative, PR-negative and HER2-negative status.	[32,33,34,35,36,37,38,39,40,41,42,43]
		Royal Brisbane Women’s Hospital [40]	Caucasian	MAGE-A: 47% (*n* = 65)	IHC	MAGE-A Antibody (6C1), Santa Cruz Biotechnology(USA)		
		Affiliated Tumour Hospital of Xinjiang Medical University [41]	Asian	MAGE-C: 38.2% (*n* = 110)	IHC	Rabbit polyclonal MAGE-C2 Antibody, Sigma-Aldrich (USA)		
		Centre of Breast Cancer of The Fourth Hospital of Hebei Medical University (Shijiazhuang Hebei) [42]	Asian	MAGE-A: 76.5% (*n* = 17)	IHC	MAGE-A Antibody (6C1), Santa Cruz Biotechnology(USA)		
		University Hospital Center Zagreb [39]	Caucasian	MAGE-A: 85.7% (*n* = 49)	IHC	3DA3 Monoclonal Antibody		
		Split University Hospital Centre, Croatia [44]	Caucasian	MAGE-A1 Specific: 69.2% (*n* = 81)	IHC	Monoclonal Antibody 77B		
				Multi-MAGE: 58% (*n* = 81)	IHC	Monoclonal Antibody 57B		
				MAGE-A10: 16% (*n* = 81)	IHC	Monoclonal Antibody 3GA11		
		European Institute of Oncology (Milan, Italy) [33]	Caucasian	MAGE-A: 32% (*n* = 50)	IHC	Antibody cocktail of monoclonal antibodies 6C1, MA454, M3H67 and 57B		
		Copenhagen University Hospital [45]	Caucasian	MAGE-A: 33% (*n* = 78)	IHC	Rabbit polyclonal anti-peptide antibody EP101638 (rab Ab 1982) raised against Mage-4, Eurogentec (Belgium)		
		National Cancer Institute (Milan, Italy) [43]	Caucasian	MAGE-A: 85.7–93% (*n* = 21)	IHC	MAGE-A3 (Clone 60054-1-Ig) Monoclonal Antibody, Proteinthec (USA)		
Mesothelin (MSLN)	GPI-anchored membrane protein	Perelman School of Medicine, University of Pennsylvania [46]	Caucasian	67% (*n* = 99)	IHC	Mesothelin Monoclonal Antibody (clone 5B2), Thermo Scientific (USA)	MSLN is significantly expressed in TNBC compared to non-TNBC and is an independent prognostic marker associated with distant metastasis and worse survival.	[46,47,48]
		University of Texas MD Anderson Cancer Center [48]	Caucasian	34% (*n* = 109)	IHC	Mesothelin Monoclonal Antibody (clone 5B2), Novocastra (USA)		
Prostate stem cell antigen (PSCA)	GPI-anchored membrane protein	University Hospital of Dresden, Germany [49]	Caucasian	17% (*n* = 90)	IHC	PSCA antibody MB1	Distribution of PSCA expression among TNBC was comparable to the total population. Patients with PSCA-positive invasive micropapillary carcinoma (IMPC) of the breast had decreased disease-free survival.	[49,50]
Receptor tyrosine kinase-like orphan receptor 2 (ROR2)	Tyrosine kinase receptor family	University of New South Wales [51]	Caucasian	87% (*n* = 295, breast cancer including triple- negative)	IHC	Human ROR2 polyclonal antibody, Sigma-Aldrich (Australia)	Breast cancer patients including TNBC expressing ROR2 had significantly worse prognoses with shorter overall survival compared to those lacking ROR2.	[51]
Sperm protein associated with the nucleus X-linked (SPANX)	Sperm function	University of Texas Health Science Center [52]	Caucasian	73% (*n* = 15)	IHC	SPANXB1 (#H00728695), Abnova (Taiwan)	SPANXB1 was frequently overexpressed in human primary and metastatic TNBC. In ER-negative patients, elevated SPANX-A/C/D was correlated with shorter distant metastasis-free survival time.	[52,53]
CTAs associated with better prognosis in TNBC								
New York oesophageal squamous cell carcinoma-1 (NY-ESO-1)	Unknown; might be involved in cell cycle progression and growth	New York Presbyterian Hospital-Weill Cornell Medical Center and UCSF Medical Center [35]	Caucasian	19.2% (*n* = 50)	IHC	NY-ESO-1 Monoclonal Antibody(E978) produced in author’s laboratory	Higher expression of NY-ESO-1 was detected in TNBC. NY-ESO-1 expression was correlated with tumour-infiltrating lymphocytes and associated with good prognosis.	[33,35,39,40,43,54,55,56,57]
		University Hospital Center Zagreb [39]	Caucasian	10% (*n* = 50)	IHC	NY-ESO-1 Monoclonal Antibody (B9.8.1.1)		
		Roswell Park Cancer Institute [54]	Caucasian	16% (*n* = 168)	IHC	NY-ESO-1 Mouse Monoclonal, Zymed/Invitrogen (USA)		
		Asan Medical Centre, Korea [56]	Asian	9.3% (*n* = 172)	IHC	NY-ESO-1 Monoclonal Antibody (E978), Invitrogen (USA)		
		Royal Brisbane Women’s hospital [40]	Caucasian	~20% (*n* = 65)	IHC	NY-ESO-1 Antibody (E978), Santa Cruz Biotechnology(USA)		
		National Cancer Institute (Milan, Italy) [43]	Caucasian	28.6% (*n* = 21)	IHC	NY-ESO-1 Monoclonal Antibody (E978), Invitrogen (USA)		
		European Institute of Oncology (Milan, Italy) [57]	Caucasian	16% (*n* = 50)	IHC	NY-ESO-1 Monoclonal antibody (E978) provided by Ludwig Institute for Cancer Research		
CTAs with oncogenic potential								
Melanoma antigen gene (MAGE)	Not known. May promote tumourigenesis and metastasis.	See Above					Promote tumourigenesis and metastasis via various mechanisms such as acting as master regulator of E3 RING ubiquitin ligase, inhibiting p53 tumour suppressor or by enhancing cell motility.	[33,34,35,36,37,38,39]
New York oesophageal squamous cell carcinoma-1 (NY-ESO-1)	Unknown; might be involved in cell cycle progression and growth	See Above					Might be involved in cellular proliferation and growth.	[21]
Preferentially expressed antigen of melanoma (PRAME)	Membrane-bound protein	National Cancer Institute (Milan, Italy) [43]	Caucasian	85.7–96.6% (*n* = 21)	IHC	PRAME Polyclonal Antibody (Clone NBP1-85418), Novus Boilogicals (USA)	Role in EMT reprogramming. Expression of PRAME was associated with negative ER status.	[58,59,60]
Sperm-associated antigen 9 (SPAG9)	Sperm function	National Institute of Immunology, Aruna Asaf Ali Marg, (New Delhi, India) [61]	Asian	NA	IHC	Polyclonal antibody to SPAG9 was prepared in authors’ laboratory	Analysis of 100 breast cancer tissues (94 infiltrating ductal carcinomas [IDC], 2 ductal carcinomas in situ [DCIS] and 4 invasive lobular carcinomas [ILC]) revealed that 88% of samples stained positive for SPAG9. Role in invasiveness of breast cancer. Downregulation could reduce invasive potential of TNBC.	[61,62]
Sperm protein associated with the nucleus X-linked (SPANX)	Sperm function	See Above					Required for metastasis. Interacts with lamin A/C at the inner nuclear membrane and involved in the formation of actin-rich cellular protrusions that reorganise the extracellular matrix.	[52,53]
Testes-specific protease 50 (TSP50)	Oncogene	Northeast Normal University (Changchun, China)	Caucasian	NA	IHC	TSP50 Monoclonal Antibody was prepared in authors’ laboratory	Analysis of 88 clinical breast cancer tissue microarrays (BR955 and BR 1101 from US Biomax, Rockville, MD, USA) revealed that 90.9% of specimens stained positive for TSP50 compared to 10% of adjacent normal tissues. Role in cell growth. Knockdown of TSP50 in breast cancer cells significantly inhibits cellular proliferation. TSP50-positive tumours were associated with negative ER expression and higher grade.	[63,64]
Zinc-finger protein 165 (ZNF165)	Gene regulation	Simmons Comprehensive Cancer Center, UT-Southwestern Medical Center, Dallas [27]	Caucasian	90% (*n* = 10)	IHC	ZNF165 (H00007718), Novus Biologicals (USA)	Enhances growth and survival of human TNBC cells both in vitro and in vivo by regulating TGF-β signalling. Frequently overexpressed in TNBC.	[27,65]
Tripartite motif containing 27 (TRIM27)	Gene regulation	Simmons Comprehensive Cancer Center, UT-Southwestern Medical Center, Dallas [27]	Caucasian	NA	TCGA		TRIM27 expression was significantly elevated in TNBC compared to normal breast tissue based on TCGA data. Displayed difference in cellular localisation, as it was mainly cytoplasmic in normal breast epithelia and more nuclear in TNBC tissues. Regulates TGFβ-dependent transcription in complex with ZNF165, ZNF446 and SMAD in TNBC.	[27,65]
Other CTAs with increased expression in TNBC								
Actin like 8 (ACTL8)	Cellular architecture	National Centre for Tumour Diseases (Heidelberg, Germany) [66]	Caucasian	57% (*n* = 98, TCGA)	TCGA		Frequently expressed in TNBC based on in silico analysis.	[66]
Chromosome X open reading frame 6/ mastermind-like domain containing 1/Kita-Kyu-Shu lung cancer antigen-1 (CXorf6/MAMDL1/KK-LC-1/CT83)	Development of male genitalia Not known	Johannes Gutenberg-University (Mainz, Germany) [67]	Caucasian	64.7% (*n* = 17, from commercial vendor)	IHC	Anti-CXorf61-A polyclonal antibody	Frequently expressed in TNBC.	[67]
		Kitasato University Medical Center (Japan) [68]	Asian	100% (*n* = 8)	IHC	Mouse monoclonal antibody was prepared by CLEA Japan (Japan)	Frequently expressed in TNBC based on in silico analysis. Frequently overexpressed in TNBC and tumours without ER expression.	[66,68]
Sperm protein 17 (SP17)	Sperm function	University of Texas MD Anderson Cancer [69]	Caucasian	47.2% (*n* = 36)	IHC	Antibody against SP17	SP17 is frequently expressed in primary breast tumours and in TNBC.	[69]
Wilms tumour-1 (WT-1)	Transcription factor	European Institute of Oncology (Milan, Italy) [57]	Caucasian	54% (*n* = 27)	IHC	WT1 Monoclonal Antibody (Clone WT49), Monosan (Netherlands)	Highest expression in TNBC compared to other breast cancer subtypes.	[57]

**Table 2 cancers-13-03875-t002:** Summary of clinical trials involving CTAs with a focus on breast cancer.

Target	Clinical Trials.Gov Identifier	Type	Drug Details	Phase	Recruitment Status	Breast Cancer Subtypes/Other Cancers
NY-ESO-1	NCT03093350	T-cell immunotherapy	Tumour-associated antigen (TAA)-specific cytotoxic T- lymphocytes targeting NY-ESO-1, MAGEA4, PRAME, survivin and SSX2	Phase II	Active, not recruiting	Metastatic or locally recurrent unresectable breast cancer
	NCT02015416	Cancer vaccine	IDC-G305: immunotherapy consisting of recombinant NY-ESO-1 antigen and the adjuvant GLA-SE	Phase I	Completed	Breast cancer, melanoma, ovarian cancer, sarcoma or NSCLC
	NCT01522820	Cancer vaccine	DEC-205/NY-ESO-1 fusion protein CDX-1401 with and without sirolimus	Phase I	Completed	Breast cancer, other solid tumours
	NCT00291473	Cancer vaccine	Cholesterol-bearing hydrophobized pullulan HER2 protein 146 (CHP-HER2) and NY-ESO-1 protein (CHP-NY-ESO-1) in combination with OK-432	Phase I	Completed	HER2- and/or NY-ESO-1-expressing cancers
	NCT01967823	T-cell immunotherapy	Anti-ESO mTCR-engineered peripheral blood lymphocytes with high-dose aldesleukin	Phase II	Completed	Metastatic cancer, including melanoma whose tumours express the ESO antigen
	NCT02661100	Cancer vaccine	CDX-1401: human monoclonal antibody specific for DEC-205, fused to full-length tumour antigen NY-ESO-1 in combination with poly-ICLC and pembrolizumab	Phase I/II	Withdrawn (Drug unavailable)	Advanced TNBC, NSCLC, small-cell lung cancer, urothelial cancer, mesothelioma, malignant melanoma
	NCT02457650	T-cell immunotherapy	Anti-NY-ESO-1 TCR transduced T cells	Phase I	Unknown	Breast cancer, other solid tumours
	NCT00623831	Cancer vaccine	mixed bacteria vaccine in patients with tumours expressing NY-ESO-1 antigen	Phase I	Completed	Breast cancer, other solid tumours
	NCT03159585	T-cell immunotherapy	TAEST16001: NY-ESO-1-specific TCR affinity enhancing specific T-cell therapy	Phase I	Completed	Breast cancer stage IV, other advanced solid tumours
	NCT01234012	Cancer vaccine	MF-001: CHP-NY-ESO-1 complex consisting of recombinant NY-ESO-1 protein and cholesteryl hydrophobized pullulan (CHP)	Phase I	Completed	Metastatic or refractory breast cancer, other solid tumours
	NCT00948961	Cancer vaccine	CDX-1401 with immune stimulants such as resiquimod and poly-ICLC (Hiltonol)	Phase I/II	Completed	Advanced malignancies expressing NY-ESO-1
MAGE	NCT04639245	T-cell immunotherapy	Genetically engineered cells (MAGE-A1-specific T-cell receptor-transduced autologous T cells) and atezolizumab	Phase I/II	Not yet recruiting	Metastatic TNBC, urothelial cancer or NSCLC
	NCT02153905	T-cell immunotherapy	Autologous T cells transduced with an anti-MAGE-A3 HLA-A*01-restricted TCR (MAGE-A3-01) TCR and aldesleukin	Phase I/II	Terminated	Breast cancer, cervical cancer, renal cancer, melanoma, bladder cancer
	NCT02111850	T-cell immunotherapy	HLA-DP0401/0402 restricted anti-MAGE-A3 TCR-gene engineered lymphocytes and aldesleukin	Phase I/II	Active, not recruiting	Breast cancer, cervical cancer, renal cancer, urothelial cancer, melanoma
	NCT00020267	Cancer vaccine	MAGE-12 peptide vaccine	Phase I	Completed	Refractory metastatic cancer expressing MAGE-12 antigen: Breast cancer, other solid tumours
	NCT03093350	T-cell immunotherapy	Tumour-associated antigen (TAA)-specific cytotoxic T- lymphocytes targeting NY-ESO-1, MAGEA4, PRAME, survivin and SSX2	Phase II	Active, not recruiting	Any breast cancer patient with metastatic or locally recurrent unresectable breast cancer
PRAME	NCT03093350	T-cell immunotherapy	Tumour-associated antigen (TAA)-specific cytotoxic T-lymphocytes targeting NY-ESO-1, MAGEA4, PRAME, survivin and SSX2	Phase II	Active, not recruiting	Any breast cancer patient with metastatic or locally recurrent unresectable breast cancer
	NCT00423254	Cancer vaccine	DNA vector pPRA-PSM with synthetic peptides, E-PRA and E-PSM	Phase I	Completed	Advanced solid malignancies: Breast cancer, other solid tumours
MSLN	NCT02792114	T-cell immunotherapy	Mesothelin-specific chimeric antigen receptor-positive T cells	Phase I	Recruiting	Breast cancer, metastatic HER2-negative breast cancer
	NCT02414269	T-cell immunotherapy	Mesothelin-targeted T cells	Phase I/II	Recruiting	Breast cancer, malignant pleural disease, mesothelioma, metastases, lung cancer
	NCT02580747	T-cell immunotherapy	Anti-meso-CAR vector transduced T cells	Phase I	Unknown	TNBC, other mesothelin-positive tumours
	NCT03102320	Antibody-drug conjugate	Anetumab–ravtansine: mesothelin-targeting antibody-drug conjugate	Phase Ib	Active, not recruiting	TNBC, cholangiocarcinoma, adenocarcinoma of the pancreas, NSCLC, gastric adenocarcinoma
	NCT02485119	Antibody-drug conjugate	BAY94-9343: anetumab– ravtansine	Phase I	Completed	Advanced malignancies
PSCA	NCT03927573	Antibody	GEM3PSCA: PSCA-targeted bispecific antibody engaging T cells	Phase I	Recruiting	PSCA-positive cancer: urogenital tract (renal, transitional cell, prostate), NSCLC, breast and pancreatic cancer refractory to standard treatments
ROR2	NCT03504488	Antibody-drug conjugate	Conditionally active biologic (CAB) ROR2-targeted antibody-drug conjugate (CAB-ROR2-ADC)	Phase I/II	Recruiting	TNBC, locally advanced unresectable or metastatic solid tumours that have failed all available standard therapies, NSCLC, soft tissue sarcoma
WT1	NCT01220128	Cancer vaccine	GSK2302024A: recombinant WT1 antigen-specific cancer immunotherapeutic (ASCI)	Phase II	Terminated (negative phase III of another study product from same technology platform)	WT1-positive stage II or III breast cancer
	NCT02018458	Cancer vaccine	Cyclin B1/WT-1/CEF (antigen)-loaded dendritic cell vaccination with preoperative chemotherapy	Phase I/II	Completed	TNBC, ER+/HER2-breast cancer
	NCT03761914	Cancer vaccine	Galinpepimut-S: WT1 analogue peptide vaccine	Phase I/II	Recruiting	TNBC, acute myelogenous leukaemia, ovarian cancer, colorectal cancer, small-cell lung cancer
	NCT01291420	Cancer vaccine	Autologous WT1 mRNA-transfected dendritic cell vaccine	Phase I/II	Unknown	Breast cancers, glioblastoma, renal cell carcinoma, sarcomas, malignant mesothelioma, colorectal tumours

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
