# Peer review of "Cancer-Testis Antigens in Triple-Negative Breast Cancer: Role and Potential Utility in Clinical Practice"

_cancers, 2021, doi:10.3390/cancers13153875_

Round 1
Reviewer 1 Report
Because the evidences are not enough, this manuscript does not reach the threshold of acceptability for publication in Cancers. |
Reviewer 2 Report
The authors have put together a nice review to summarize current knowledge on cancer-testis antigens in triple-negative breast cancers.
I enjoyed reading, and believe this can help other authors who are studying breast cancers (TNBC). Certainly, it has the potential to help facilitate less investigated areas of research using the CTA.
However, there is one critical area that can improve the manuscript, as well as the translational potential of this research.
As shown in the staining examples, it appears that CTA markers are tested either by IHC or multiplex IHC that use multi-colored labels.
1) In current clinical labs, it is imperative to use a standardized method of measurement, using genomics, or immunohistochemistry based. I understand this might be slightly out of scope for this paper. However, it is critical for future translation and adoption by other investigators in the field.
Therefore, I suggest that at least the type of assay and antibody each lab has used - as two additional column.
2) then, it would be great to know - in antigens that were tested as part of multiplex IHC, or IF - if these antibodies can be further developed in CLIA environment to have clinical translational potential. Or would authors propose to develop a multiplex staining platform as a solution? So far, in the clinic, it is hard to conduct a clinical quality multiplex staining - therefore, difficult to translate. I think this part is critical since the conclusion of the paper is future translation, including the clinical treatment development.
3) Lastly, is there any data to support that inhibiting these CTAs can have therapeutic potentials at all?
Once we can add this info, it would be ready to go forward !!
Reviewer 3 Report
the authors submitted an interesting and comprehensive review on the topic. Discussion is well articulated, and perspectives outline.
Author Response
Thank you for your comments.
Round 2
Reviewer 1 Report
This paper is accepted.